# Selenium Biofortification: Roles, Mechanisms, Responses and Prospects

**DOI:** 10.3390/molecules26040881

**Published:** 2021-02-07

**Authors:** Akbar Hossain, Milan Skalicky, Marian Brestic, Sagar Maitra, Sukamal Sarkar, Zahoor Ahmad, Hindu Vemuri, Sourav Garai, Mousumi Mondal, Rajan Bhatt, Pardeep Kumar, Pradipta Banerjee, Saikat Saha, Tofazzal Islam, Alison M. Laing

**Affiliations:** 1Bangladesh Wheat and Maize Research Institute, Dinajpur 5200, Bangladesh; 2Department of Botany and Plant Physiology, Faculty of Agrobiology, Food and Natural Resources, Czech University of Life Sciences Prague, Kamycka 129, 165 00 Prague, Czech Republic; 3Department of Plant Physiology, Slovak University of Agriculture, Nitra, Tr. A. Hlinku 2, 949 01 Nitra, Slovakia; 4Department of Agronomy, Centurion University of Technology and Management, Paralakhemundi 761211, India; sagar.maitra@cutm.ac.in; 5Department of Agronomy, Bidhan Chandra Krishi Viswavidyalaya, Nadia, West Bengal 741252, India; sukamalsarkarc@yahoo.com (S.S.); garai.sourav93@gmail.com (S.G.); mou.mousumi98@gmail.com (M.M.); 6Department of Life Sciences, The Islamia University of Bahawalpur, Bahawalpur 58421, Pakistan; zahoorahmadbwp@gmail.com; 7International Maize and Wheat Improvement Center, Patancheru, Hyderabad 502324, India; hinduvemuri@gmail.com; 8Regional Research Station, Kapurthala, Punjab Agricultural University, Ludhiana, Punjab 144601, India; rajansoils@pau.edu; 9Agronomy (Crop Nutrition) DES (Agronomy) FASC (Farm Advisory Service centre) Extension Centre of PAU, Ludhiana Posted as District Incharge at Kapurthala, Punjab 144601, India; pardeep.agron10@gmail.com; 10Department of Biochemistry and Plant Physiology, Centurion University of Technology and Management, Paralakhemundi 761211, India; pradipta.banerjee@cutm.ac.in; 11Subject Matter Specialist (Agricultural Extension), Nadia Krishi Vigyan Kendra, Bidhan Chandra Krishi Viswavidyalaya, Gayeshpur, Nadia, West Bengal 741234, India; saikatsaha2012@gmail.com; 12Institute of Biotechnology and Genetic Engineering (IBGE), Bangabandhu Sheikh Mujibur Rahman Agricultural University Gazipur, Gazipur 1706, Bangladesh; tofazzalislam@yahoo.com; 13CSIRO Agriculture and Food, 4067 Brisbane, Australia; alison.laing@csiro.au

**Keywords:** selenium, trace element, nutrition, humans, animals, plants, biofortification

## Abstract

The trace element selenium (Se) is a crucial element for many living organisms, including soil microorganisms, plants and animals, including humans. Generally, in Nature Se is taken up in the living cells of microorganisms, plants, animals and humans in several inorganic forms such as selenate, selenite, elemental Se and selenide. These forms are converted to organic forms by biological process, mostly as the two selenoamino acids selenocysteine (SeCys) and selenomethionine (SeMet). The biological systems of plants, animals and humans can fix these amino acids into Se-containing proteins by a modest replacement of methionine with SeMet. While the form SeCys is usually present in the active site of enzymes, which is essential for catalytic activity. Within human cells, organic forms of Se are significant for the accurate functioning of the immune and reproductive systems, the thyroid and the brain, and to enzyme activity within cells. Humans ingest Se through plant and animal foods rich in the element. The concentration of Se in foodstuffs depends on the presence of available forms of Se in soils and its uptake and accumulation by plants and herbivorous animals. Therefore, improving the availability of Se to plants is, therefore, a potential pathway to overcoming human Se deficiencies. Among these prospective pathways, the Se-biofortification of plants has already been established as a pioneering approach for producing Se-enriched agricultural products. To achieve this desirable aim of Se-biofortification, molecular breeding and genetic engineering in combination with novel agronomic and edaphic management approaches should be combined. This current review summarizes the roles, responses, prospects and mechanisms of Se in human nutrition. It also elaborates how biofortification is a plausible approach to resolving Se-deficiency in humans and other animals.

## 1. Introduction

Selenium (Se) is a micronutrient essential for the proper functioning of plants and animals [1,2]. It was first described by the Swedish chemist Jacob Berzelius in 1817, however, its biological role was not determined until the 1950s when overdoses were linked to cardiac muscle dystrophy and acute hepatic necrosis [3]. In 1973, the beneficial biological role of Se as a key constituent of glutathione peroxidase (GPx) was discovered; thus it contributes to protecting the body from stress-induced oxidative damage to living cells. Subsequently, it was found that Se is not only associated with GPx, it is also connected with numerous other enzymatic activities within organisms’ cells; for example, Se biological forms like SeCys and SeMet are key constituents of iodothyronine deiodinase, which is crucial in the healthy functioning of the endocrine system [1,4].

It is estimated that globally one billion people are facing Se insufficiency [5] since Se is essential for the good functioning of the human immune, endocrine and reproductive systems, and also links with the function of the human brain [6]. An earlier study revealed that prolonged Se deficiency in the human body negatively affects the cardiovascular system and can lead to myocardial infarction, i.e., heart attacks [7]. Besides these examples, Se deficiency is also associated with Keshan and Kashin-Beck diseases, which predominantly occur in children and women of child-bearing age [8,9].

Generally, Se is uptaken by humans through consumption of Se-enriched plant and animal products, in particular from plant sources. Plants take up Se from the soil as selenates [10], which are then converted into the organic forms SeCys and SeMet [11]. Increasing the concentration of Se in edible plants’ parts/products is a viable pathway to overcome human Se deficiency. Recently, Se-biofortification combined with improved agronomic, breeding, molecular, biotechnology, and genetic engineering approaches have been recognized as a leading tool to enrich plant-food products with Se. While there is a large, disperse body of literature on the roles and mechanism of Se in human health and its enrichment in plant foods, a comprehensive review covering all these aspects is lacking. This review summarizes the roles, responses, and mechanisms of Se in humans and the potential to enrich plant foods with this micronutrient through biofortification. The review also explains how biofortification is a plausible approach to resolving Se-deficiency in humans and other animals.

## 2. The Natural Form of Selenium and Its Deficiency and Toxicity Symptoms

### 2.1. Natural Form

Selenium (Se) is widespread across the Earth and is present in the atmosphere, lithosphere, hydrosphere and biosphere [12]. The weathering of rocks and the eruption of volcanic gases are key sources of Se into the environment. Additionally, the decomposition of Se-enriched organic matter, through biomethylation by microorganisms, maintains a positive flow of this element into the atmosphere [1]. These mechanisms contribute to the presence of volatile Se compounds, viz., hydrogen selenide (H_2_Se), dimethyl selenide (DMSe), and selenium oxide (SeO_2_). Globally, the Se content in arable soils ranges between 0.33 and 2 mg/kg [13]. Se-rich areas are known as seleniferous areas [14]. The Se concentration in soils varies depending on the management of the local environment and the presence of both Se-rich parent materials and microorganisms necessary for its release into the atmosphere [15].

The soils which have originated from igneous rock, granite, sandstone, and limestone are all rich in Se [16]. Conversely, soils in temperate and humid climates are generally low in Se. Irrespective of soil depth, the Se content in mineral-enriched soils fluctuates ~14 mg/kg [17]. Se-rich materials include berzelianite (Cu_2_Se), klaustalite (PbSe), and naumanite (Ag_2_Se) [18]. Anthropogenic activities, such as the combustion of fossil fuels, metal smelting, international shipping and the over-use of inorganic fertilizers are primarily responsible for additional contributions of Se into the atmosphere (and thus into agricultural soils), from the before mentioned minerals [17].

Se-rich soils have been observed in the United States, Russia, parts of China, Australia, Canada, and Ireland [4]. In contrast, New Zealand and a wide portion of Europe have soils which are largely Se deficient [1]. Tomza-Marciniak [19] calculated that Se deficiencies exist in more than 70% of countries. However, the total Se content in soil is not a reliable estimate of plant available Se or the amount of Se available to humans and animals through plants [20]. The Se availability of plants is determined by a large number of soil chemical and biochemical characteristics, including sorption, disruption, soil pH, presence of other nutrients and methylation [21]. For example, Se uptake is higher for plants high in sulfur [22]. Se-rich foods include seafood, eggs, chicken, nuts, mushrooms, and green vegetables including spinach, cauliflower, and cabbage. Se concentrations in humans vary with agroclimatic region and daily diet.

In water, Se is present in minute quantities as selenates or selenites. Groundwater contains higher concentrations of Se than seawater [16,18]. This is largely the result of runoff of Se-rich fertilizer from intensively managed agricultural soils, as well as Se secretion from parent rock material [23]. In potable water, 10 µg Se per litre of water is acceptable according to the World Health Organization [24].

### 2.2. Selenium Deficiency Symptoms

#### 2.2.1. Symptoms in Human

Prolonged Se deficiency adversely affects the cardiovascular system, which may be a cause of myocardial infarction [16]. Se deficiency also causes Keshan and Kashin-Beck diseases, which primarily occur in childbearing women and in children who live in areas deficient in Se [25]. A moderate deficiency of Se in daily food habit reduces immunity, can impair the nervous system and may cause congenital hypothyroidism in fetuses [26]. Additionally, Alzheimer’s disease, depression and anxiety have been associated with prolonged Se deficiencies [27]. Se may contribute to the suppression of HIV and slow the progression to AIDS [28]. Se is also necessary for the developing fetus in women and animals [16,29]. An insufficiency of Se in the human diet affects the thylakoid gland and may lead to moodiness and the impairment of behaviors and cognitive functions [30,31,32]. Se deficiency reduces the activity of 5′-thyronine deiodinase enzymes, leading to low triiodothyronine concentration in blood. Moreover, a deficiency of Se accelerates human ageing [31].

#### 2.2.2. Symptoms in Animals

The majority of Se-deficiency symptoms have been observed in animals when less than 0.1 mg Se is present per kg of animal diet. Most commonly, Se deficiency causes a myo-degenerative syndrome [32] Table 1, also known as white muscle disease (WMD), whereby animal muscles are pale in appearance [33]. WMD may occur in all livestock including birds, and most seriously damages the skeletal (fibres/highly elongated cells) and cardiac muscles [34]. The deficiency of Se also impairs an animal’s immunity [35]. The clinical signs of Se deficiency in animals include reductions in appetite, fertility and growth, as well as muscle weakness [36]. Specific Se deficiency symptoms include heart disease in pigs, placenta retention in cows, and birds both a higher embryonic mortality rate and muscular dystrophy [33,37].

### 2.3. Selenium Toxicity

#### 2.3.1. Toxicity in Humans and Animals

Both excessive and insufficiency of Se are detrimental to human health. The consumption of higher doses of Se can be toxic. There is a narrow limit between safe-and-adequate Se uptake and overconsumption leading to toxicity. There are only limited reports on human Se toxicity; this may be because many commonly consumed foods are Se non-accumulators. The symptoms of Se toxicity are hair loss and skin and nail lesions [38], hypotension, tachycardia, muscle contractions dizziness, nausea, vomiting, facial flushing, tremors and muscle soreness. In extreme cases, acute Se toxicity can cause serious intestinal and neurological problems, heart attack, kidney failure and death [39]. Excessive Se uptake can also damage the mucus membranes of the digestive tract, and lead to ongoing nausea, diarrhoea and also increase risk of type 2 diabetes [38]. In the case of animals, the death of poultry due to Se toxicity in wheat grains used as chicken feed was reported in South Dakota, USA [40].

The limit between safe and toxic amounts of Se is small and has yet to be standardized in many geographical locations. The recommended dietary allowance of Se varies in humans with age, gender, pregnancy and lactation Figure 1. Pregnant or lactating women require higher 9% and 27% daily amounts of Se than other women [41]. Human beings must consume around 55 micrograms of Se per day and not exceed the maximum limit of 400 micrograms per day [42]. The World Health Organization recommends a daily average intake of 55 µg of Se, while the recommendation is varied with age, gender, diet and geographic location [43]. The International Food and Nutrition Board has recommended an average daily intake of 40–70 µg Se and 45–55 µg Se for men and women, respectively, and 25 µg Se for children [2,44]. A daily dose of 55–200 µg of Se is recommended for healthy adult humans [45,46]. Table 2 illustrates the maximum recommended daily Se intake limits, above which symptoms of Se-toxicity such as significant hair loss and abnormal nail growth appear [41]. 

In case of the available forms Se, Ríos et al. [47] regarded that >40 μmol Se/L of selenate as toxic, while Hawrylak-Nowak [48] estimates that safe concentrations of Se are between 20 and 15 μmol/L for selenate and selenite, respectively. However, Ríos et al. [49] reconsidered and set an upper limit of 80 μmol Se/L.

#### 2.3.2. Selenium Phytotoxicity

Toxic levels of Se within plant tissues are greater than 5 mg/kg [50]. Se toxicity does not occur when selenate is applied at rates between 10–200 g Se per ha, which are commonly recommended for wheat biofortification. Se is usually not considered essential for taller plants, while soils low in Se are considered to neither inhibit plant growth nor reduce crop yield [50,51]. However, some research has indicated the beneficial effects of low doses of applied Se on crop performance. For example, increased growth of ryegrass (*Lolium perenne*) and lettuce (*Latuca sativa*) was recorded when crops were fertilized with Se and exposed to UVB radiation. Low-dose applications of Se are useful for crop plants only when they are under oxidative stress, otherwise, this microelement is not essential.

## 3. Importance of Selenium for Global Human Nutritional Security in the 21st Century

Se is an essential mineral element, obtained by humans and animals through their diets. Biofortification is a plausible strategy to enrich Se in foods. Options to enhance Se content in human foods include: increasing the number of foods naturally high in Se (e.g., Brazil nuts); agronomic biofortification (i.e., incorporation of Se fertilizers); genetic biofortification; plant breeding strategies; and supplementing organic Se directly to humans or livestock [52,53]. Those living in areas with low endemic levels of Se generally do not consume adequate amounts of Se, as food crops and animal feeds do not take sufficient Se from the soils in which they grow. In the case of plants, the amount of Se in plants depends on the Se concentration in the soils in which the plants are grown. A study of Se concentration in Brazil nuts demonstrated high spatial variability: while a single Brazil nut from one area contained 288% of the recommended daily Se intake, other nuts are grown elsewhere contained only 11% of the recommended daily Se intake [42].

### 3.1. Health Benefits of Selenium for Humans

Among the total amount of Se in an average human (~3–20 mg), around 47% is found in skeletal muscles/fibre cells and about 4% in the kidneys [54]. Generally, the presence of Se is determined by its concentration in blood serum [55]. Highest Se concentrations are observed in adults around 60 years of age after this Se concentration gradually decreases [56]. A Se deficiency occurs when the blood contains less than 85 µg Se per litre of blood serum. Low Se levels increase the risk of some cancers [57] while over-sufficiency of Se may result in anaemia, hair loss, bone stiffness or blindness [54]. Where Se is present in the air in concentrations above 0.2 mg/m^3^ it may be inhaled by humans and other animals [58]. Both under- and over-supply of Se are harmful to human health [59].

Daily doses of 100–200 µg Se inhabits of genetic damage and the development of some cancer cells [60]. A daily intake of over 200 µg Se per day may accelerate the presence of cytotoxic T cells within the body. A daily intake of 100 µg Se has been observed to reduce clinical depression and anxiety [61]. Exceeding an intake of 1500 µg day^−1^ has been observed to be toxic to humans [62]. As the Se concentration in plant and animal foods reflects the endemic Se concentration of the soil in which they are grown, Se concentration in foods varies globally. For example, Se intake is more than 90 µg/day in the United States [59], in Venezuela, it is 326 µg/day [63], while in some European countries the concentration is lower than the recommended value at around 30 µg/day [64]. The Health benefits of Se in the physiological processes of the human body are illustrated in Figure 2.

The production of Se-enriched foods, such as eggs, meat, and milk, is attractive worldwide as these products improve the nutritional status and health of those who may otherwise be Se deficient. However, while this food biofortification would benefit those with Se deficiencies, people with high inherent Se intake through their diet may be adversely affected by the biofortification process. Therefore, it is recommended that people living in areas with high Se concentrations in the soil do not consume excessive amounts of Se-fortified foods [65]. Se is a powerful antioxidant which may contribute to fighting cancers, viral infections, and aging. It is important for the normal functioning of the thyroid, brain, heart, and reproductive system. Se is the only micronutrient (as SeCys protein) mentioned in the human genome [66]. The effect of Se in mitigating some cancers has been reported by Combs and Lu [67]. The numerous health benefits of Se are discussed in the following sub-sections.

#### 3.1.1. Selenium is a Strong Antioxidant

Se acts as a strong antioxidant, which reduces cellular damage caused by free radicals and protects the body from heart disease and some cancers [68] Figure 2. Free radicals are naturally produced as byproducts of living cells and may also be a result of smoking, drinking or mental stress [69]. Furthermore, the production of free radicals has been linked to many human diseases including cardiovascular disease, Alzheimer’s, some cancers, and premature ageing [70].

#### 3.1.2. Selenium Reduces the Risk of Some Cancers

Se reduces the risk of some cancers [71,72]. Se reduces DNA damage and the effects of stress while boosting the body’s immune system. The WHO recommended daily dose of Se improves the quality of life in patients undergoing chemotherapy [73]. These beneficial effects of Se have been observed in Se-enriched plant foods but not in supplements. As well, Se may reduce the side effects of chemotherapy [71,72,73].

#### 3.1.3. Selenium Protects Against Cardiovascular Problems

A Se-sufficient human diet contributes to a healthy heart and cardiovascular system [65,74]. Se reduces the risk of heart attack by mitigating inflammation and oxidative stress in the body which otherwise may contribute to plaque build-up within the lining of arterial walls, leading to atherosclerosis [75]. Arterial thickening contributes to heart attacks and heart disease.

#### 3.1.4. Selenium May Improve Some Mental Illnesses

Insomnia may contribute to anxiety, depression and reduced quality of life [76]. Often patients with poor mental health have been found to have low blood Se levels [77].

#### 3.1.5. Selenium Is Beneficial for Thyroid Health

Selenium helps protect the thyroid against oxidative cell damage and aids the release of thyroid hormones which regulate growth and development with the human body [78,79]. Se is thus beneficial for the normal functioning of the thyroid gland [80,81].

#### 3.1.6. Se Strengthens Immunity and May Reduce Breathing Difficulties

Elevated Se levels in blood serum have been associated with an enhanced human immune system [82,83,84]. Additionally, Se may benefit some asthmatics by reducing swelling in airways [85,86].

#### 3.1.7. Finland Case Study: Selenium Biofortification of Human and Livestock Feed Crops

Heart disease was a leading cause of death in Finland in the 1960s and 1970s. To counter this, the Finnish government required the introduction of Se (as selenate) fertilizer in all multiple-nutrient fertilizer mix used for agriculture from 1984. Table 3 illustrates the success of this Se-biofortification program in Finland. Initially, Se fertilizer was applied at 16 mg Se/kg of fertilizer in grain and horticultural production and 6 mg Se/kg for pasture and hay production. The program was so successful in increasing plant Se concentrations, and thus the level of Se in humans in Finland, that in 1990 a rate of 6 mg Se/kg fertilizer was adopted universally [87,88]. In 1998, Se supplementation was increased to 10 mg Se/kg fertilizer for all fertilizers applied to crops [87]. As a result of the agronomic Se biofortification program, human health in Finland increased markedly: dietary Se intakes trebled and blood plasma Se concentrations approximately doubled within 3 years of the introduction of the biofortification [88]. The prevalence of heart disease and some cancers in Finland have significantly decreased since 1985.

### 3.2. Importance of Selenium for Both Plants and Animals

#### 3.2.1. For Plants

In plants, Se is most commonly found as SeMet, methyl-SeCys or ɣ-glutamyl-Se-SeCys (ɣ-Glu-MeSeCys) [93]. Supplementing wheat and maize with Se at rates of up to 100 gm Se/ha did not affect crop productivity in terms of grain or stover yield [88]. Additionally, applications of Se have been reported to increase plants’ resistance to oxidative stresses [94,95]. Further, the presence of the antioxidants glutathione peroxidase (GSH-Px) and superoxide dismutase (SOD) is enhanced with Se supplementation and, in some cases, the concentration of lipids are reduced. Hence, Se may enhance the natural defence mechanisms of plant against pests, particularly insects [96]. D’Amato et al. [97] reported that the oil yields from olive trees were enhanced in terms of intensity and stability of color following application of Se. Se biofortification in peach and pear trees increased the Se concentration in fruit and reduced fruit-softening rates, thus increasing the shelf-life of fruit products [98]. However, biofortification with high concentrations of Se negatively affected plant growth in lettuce [48] and retarded germination in mustard [98]. Some crops, such as brassicas (e.g., broccoli, *B. oleracea* L.) were unaffected by applications of Se [99].

Se fertilizers can be effectively utilized to alleviate poisoning from some toxic metal and metalloids stress in field crops. Dipping the roots of rice seedlings in a Se solution before transplanting reduces the effects of arsenic, which inhibits seedling growth [93,100]. Additionally, rice plant height, number of tillers, chlorophyll content, panicle length and kernel weight were also all significantly enhanced when rice seeds were coated in Se before sowing [101,102]. Se-coating rice seeds also reduce arsenic phytotoxicity seedlings and enhance crop productivity [100,101,102,103].

#### 3.2.2. For Animals

In ruminant animals, Se is more readily absorbed in organic forms than in inorganic [104]. SeMet, a commonly occurring form of Se, can be taken up in animal cells instead of methionine, as these cells do not distinguish between methionine and SeMet during protein synthesis [105]. Organic SeMet is a valuable source of Se to facilitate the rapid synthesis of seleno-proteins [106].

Se-fortified alfalfa, fed daily, is one of the most effective tools to improve Se concentration in animals [107]. This significantly increases the Se concentration in the edible meat and provides adequate Se to humans who consume the meat. The concentration of Se in an animal’s meat is directly proportional to the Se concentration in its feed or forage [108]. Marine animals take up Se from seawater, phytoplankton, and other marine feeds, such as krill [109]. Water-soluble Se is also absorbed through the gills, epidermis and gut of marine animals. There is a strong correlation between low Se levels and mercury absorption in fish [110]. Mercury toxicity reduces with higher Se concentration in fish tissues, this relationship may be important for research into aquatic food farmed under suboptimal water quality conditions.

## 4. Biofortification—A Sustainable Agricultural Strategy for Reducing Micronutrient Malnutrition

The concentration of Se in food ultimately depends on the concentration of plant-available Se in cultivated soils. The availability of Se in soils is mediated by soil pH, redox potential, cation exchange capacity (CEC) and the soil concentrations of S, Fe, Al and C [111,112]. This microelement is present everywhere but not uniformly distributed. Lyons et al. [113] illustrate the variability in soil Se concentration as shown in Table 4.

Biofortification of crops and animal feed improves the nutritional quality of food products and may help to reduce global malnutrition [88]. Plants are the most abundant sources of Se in many countries, followed by meat and seafood sources [114]. Thus, it is critical to enhancing Se uptake by plants, and ultimately the Se concentration in the human diet, to alleviate Se deficiency-induced human disorders. Se biofortification is the most effective approach to increase Se concentrations from agriculture [115].

**Table 4 molecules-26-00881-t004:** Variability in Se concentrations in soils from different locations across the globe.

Locations	Soil Types	pH (H_2_O)	Total Soil Se µg/kg	Se in Cereal Grain µg/kg	References
Yangshuo, China	Ishumiso	8.3	700	20	Lyons et al. [113] Zhu et al. [116]
Minnipa, South Australia	Calcareous Xerochrepts	8.6	80	720	Lyons et al. [113] Williams et al. [117]
Charlick, South Australia	Typic Natrixeralf	6.6	85	70	Lyons et al. [113] Thavarajah et al. [118]
East Zimbabwe	Typic Kandiustalf (ex-granitic parent material)	5.0	30.000	7	Lyons et al. [113] Winkel et al. [119] Fordyce et al. [120]

### 4.1. Selenium Biofortification through Agronomic Management

Se is a vital micronutrient element which also enhances the antioxidants properties of food crops [121,122]. Se is present in animals (including humans) and plants and influences many physiological processes including promoting growth [123]. It is believed that many populations have less than the necessary intake of Se, which increases the risk of many diseases [124]. Se enriches in plants varies depending on geography and Se availability within soils, and thus Se uptake by humans varies but is tied to diet [125]. Worldwide, the distribution of Se within soils varies topographically, with a diversified range (from near zero up to 1200 mg kg^−1^) [126]. Therefore, in many locations’ agriculture plants require Se inputs from external sources [127].

Different agronomic management practices can be used to biofortify Se in plants: these include soil inorganic fertilization, foliar application, and also as organic fertilizers. Se-biofortification through agronomic management aims to standardize and regulate the concentration of Se within crop products by incorporating the right time, right source, the right method, and the right amount of Se fertilizer. Agronomic Se bio-fortification has many advantages over direct Se supplementation as the inorganic Se absorbed by the plant is transformed into an organic form, which has a higher bioavailability [128]. The application of Se fertilizers to soil commonly results in increased total and bioavailable Se [88], and thus in higher Se concentration in edible crop products [20,129]. Agronomic Se bio-fortification reduces the risk of Se leaching into groundwater or lost through volatilization, as Se binds to soil organic matter and remains within the soil. Field tests using inorganic Se fertilizers in Finland and New Zealand successfully demonstrated its bioavailability to plants [130,131]. Foliar Se fertilization in rice, wheat, and lettuce [132] suppressed nutritional deficiencies in human populations [133].

#### 4.1.1. Selenium Biofortification through Direct Soil Fertilization with Inorganic Fertilizers

Applying selenate-based fertilizers enhance Se uptake by plants and, subsequently, the Se concentration in animals and humans who consume those plants [126]. This management practice is used where Se is in deficit by adding Se to inorganic fertilizers applied to agronomic soils, most commonly as selenate or as selenite of sodium or barium salts, which are applied to the soil either directly or diluted in a water-based fertilizer [134]. However applying Se broadly at high concentrations is generally not economically sustainable, and so site-specific Se applications which account for existing Se available within the soil and crop demand must be considered [135]. For example, Ramkissoon et al. [88] observed that application of 3.33 µg kg^−b^ of Se (equivalent to 10 g ha^−o^) to wheat can be made more efficient by its co-application with macronutrient carriers, either to the soil or to the leaves during the awn-peep stage and observed that grain Se concentrations varying from 0.13–0.84 mg kg^−w^. For raising the grain Se concentrations, soil application of selenate was found 2–15 times more effective than application of granular Se-enriched macronutrient fertilizers such as N, P, K or S. While co-application of Se applied as foliar with an N carrier doubled the Se concentration in wheat grains compared to the application of foliar Se.

In agricultural production systems, Se is taken up from the soil by plants, this must be balanced by the application of Se fertilizer. The amount of Se fertilizer applied depends on the Se concentration in both the fertilizer and the soil. Se also accumulates in soils through atmospheric deposition and from irrigation water and some industrial processes [127].

The solubility, mobility, and bioavailability of Se within a soil generally depend on its exact chemical form and how sturdily it is tied to soil particles. For example, in alkaline soils the dominant form of Se is selenate and the absorption of selenate by plants is affected by the presence (or not) of different ions which may be present in the soil (e.g., K^+^, Ca^2+^, Mg^2+^, SO_2_^–^, Cl^−^) [136,137,138]. Different fertilizers differently influence the amount of Se within a soil as they alter the concentration of different ions within the soil, and thus the availability of Se to crops. Absorption of Se within soils may be reduced due to the presence of competitive ions such as K^+^, Ca^2+^, Mg^2+^, SO^2^^−^ and Cl^−^ [139].

Se absorption is increased in soils by the addition of chelating agents such as compost and other organic matter. However, adding organic matter to soil may reduce the Se uptake capacity of crops [136]. As Se is not used in agriculture to improve soil fertility or crop productivity and thus management practices which facilitate Se bioavailability to plants are often not prioritized by farmers [123]. Therefore, it may be helpful to identify the principal factors that limit Se concentration within soils and address these to enhance the concentration of Se within the animal and human foods.

Se is well suited to agronomic biofortification of food crops. Research has shown that the selenite form (SE) of Se, when used in biofortification, is readily taken up by plants in a wide range of soils. This SE is transported throughout the plant and stored in its numerous edible parts [140]. In cereal crops, this SE is transformed into seleno-methionine and stored in the endosperm. Therefore, milled products like white flour and polished rice are significant bioavailable sources of Se [137].

Selenate (i.e., Se in its highest oxidation form, +6) is a more effective form for use in soil fertilization than selenite (i.e., Se in +4 oxidation form) [141,142]. Selenate is up to 33 times more effective than selenite in plant fertilization [126]. In soils with a high clay, content selenite is absorbed quickly but is of limited availability to plants. Applications of selenite increase grain yield in lentil by 10% with significant antioxidant activity (66% inhibition) compared to the control (59% inhibition) [143]. The form of selenates is highly water-soluble and available to uptake by plants, while easily leached from the soil solution. In aerated soils with neutral to higher pH, this form of Se is dominated. For example, soils with a high content of Ca and Mg CaSeO_4_ generally creates MgSeO_4_, this form is easily soluble and represent total Se soluble in soil [144]. In soils rich in organic matter and water and without air entry selenates are transformed and reduced to less mobile forms. With decreasing pH and redox potential in soil SeO_3_^2−^ dominate, being less available for plants than SeO_4_^2−^ [145].

#### 4.1.2. Selenium Biofortification through the Foliar Application with Inorganic Fertilizers

In some circumstances, soil-based applications of Se-enriched fertilizers are ineffective, largely due to the physical, chemical and biological soil properties, including soil texture, pH, redox potential, microbial activities and inherent Se concentration [146]. As an alternate method of fertilization, the foliar application is widely accepted [126]. The effectiveness of this method for Se supplementation has been successfully reported in various crops such as rice [147], wheat [148], potato [149], soybean [150] and carrot [151]. However, when Se concentration exceeds 100 µg Se/mL of water, it causes phytotoxicity in some crops [151]. During foliar application (SeF), the Se solution must be distributed using well-calibrated equipment; spraying should not occur on rainy or windy days, and applications need to be made at the late vegetative stage, where there is an adequate surface area to facilitate maximum absorption of Se. In the soil application (SeS), Se is effective in the period from early growth of seedlings to plant maturity for uptaking Se by roots [152].

Comparison of the long-term environmental impact of Se application directly into soils or as a foliar spray has shown that foliar application reduces the potential accumulation of Se in the environment. Lower concentrations of Se are used in foliar applications than when fertilizer is applied directly to the soil [153].

Se fertilizer is generally applied in soils with a pH 5.5 or greater. Se translocates within the plant through the phloem and is deposited in the edible portions of crop plants as SeMet. This organic form of Se is found across all cereal grains, particularly in white flour and polished rice, and is readily absorbed by the small intestine of both humans and animals [52].

At the same time, Se biofortification of agricultural soils is a relatively wasteful process. Approximately 12% of Se applied directly to soils is recovered in plants; this amount is higher when foliar applications are used [88]. For example, a field study with the purple-grained wheat cultivar (‘202w17′) and common wheat cultivar (‘Shannong 129′) showed that both soil and foliar application of Se boosted the organic Se concentration in roots, shoots, and grains of both cultivars, but the higher concentration of Se in the grain of two cultivars was noted when Se was applied as foliar. Foliar application of Se enhanced approximately 1.5-fold higher concentration of organic Se in grains of cultivar ‘202w17′ than cultivar ‘Shannong 129′ [152].

Foliar application of Se is safer and more effective, easy and cost-effective for Se biofortification than other Se fertilizer application methods [90]. Foliar application of Se at an appropriate crop growth stage accelerates the efficiency of fertilizer uptake and boosts crop production, reducing wastage. The timing of Se application is critical in cereal crops. The most effective time for Se fertilization is between booting and milking stages as this is when there is a maximum number of green leaves within the field [129].

#### 4.1.3. Selenium Biofortification through Organic Fertilizers

The presence of organic matter in drier environments increases the absorption rate of micronutrients, including Se, and may act as a reservoir of these trace elements for crops [154]. Using Se-enriched organic or green fertilizers is another option for soil amendment to produce Se-biofortified crops [155]. Well decomposed green manure facilitates Se uptake by different plant species; concurrently the decomposition of the organic matter helps to mobilize other major soil nutrients [156]. For example, when Se-enriched *Stanleya pinnata* was applied to carrots under optimum soil moisture, 90% of the organic Se was converted into inorganic selenate and selenite [156]. Furthermore, applications of SeMet and SeCys from organic sources increased Se uptake by plants which received Se containing inorganic fertilizers [157]. Se-enriched peat has been used as a source of organic Se in cucumber, tomato and lettuce crops [158]. The effectiveness of growing mushrooms in Se-enriched agricultural by-products has been demonstrated by Bhatia et al. [159]. However, Se bio-accessibility may be affected by the formation of indigestible Se -containing polysaccharides or Se association with chitin-containing structures in cell walls [160].

Applications of expensive and inefficient inorganic fertilizers are not necessary to fertilize crops Se [161]. Organic matter within the soil performs a significant role in immobilizing Se and making it available to plants [162]. The presence of multiple chelating compounds in organic matter helps enhance the uptake of Se by plants [152]. Adding organic materials like manures and crop residues to agricultural soils additionally increase the bioavailability of Se in the soil and may balance out Se levels [163]. For example, organic matter has been used to reduce Se toxicity in soils over-rich in Se [164,165]. Applying Se-loaded animal manures to the soil can result in higher Se concentrations in crops [166,167]. The organic biofortification of Se has become more widespread globally as Se-enhanced waste products have been recognized as a useful resource to improve the Se concentrations of agricultural soil and, indirectly, improve the nutritional status and health of humans [168]. Losses of Se through volatilization or leaching to groundwater in maize have been reduced by the application of soil organic matter concurrently with Se fertilizer [169].

Among these four valence states of Se, i.e., Se (0), Se (II), Se (IV) and Se (VI). The selenite Se (IV) and selenate Se (VI) forms of Se are primarily used in the biofortification of crops. Both selenite and selenate are water-soluble, with selenate more soluble than selenite in soil solutions. Under acidic conditions in humid areas (such as many tropical soils), selenite is the more commonly prevalent form of Se [170,171]; it forms strong bonds with metal oxides or soil organic matter has low bioavailability to plants [101]. However, Se, when mixed with Organo-Se complexes (for example Seleno-amino acid), exists as the valence stage of Se (II) which is exceptionally bioavailable. Se accumulation in plants is higher when applied by mixing with organic compounds than the inorganic forms of Se [157]. For example, seleno-amino acids are active compounds and can be applied to the soil in Se-amended organic manures in crop fields [172]. Besides, organo-Se compounds can also be released in soils through the decomposition of plant material and soil microbial matter [173].

### 4.2. Success of Selenium Biofortification in Food Crops Depends on a Better Understanding of the Genetic Variation of Crop Cultivars

Deep knowledge of Se biogeochemistry, its uptake mechanisms and assimilation by plants is necessary for effective Se bio-fortification of food crops. Plant-based bio-fortification is the most commonly used, most effective, natural strategy to improve deficiencies in nutritional elements like Se in the staple crops across the world [128]. Se in plants is metabolized into plant tissues [174] and varies significantly by plant species and variety [175]. Significant genetic variation in grain Se concentration has been reported for several kinds of cereals including bread wheat [176], barley [177], oat [178] and rice [179]. Similarly, significant genetic variation in seed Se concentration has been observed across legumes including chickpea, lentil [180], mung bean [181] and soybean [182]. Significant genetic variation has also been reported for Se concentrations in various leafy vegetables including onions [183], brassicas [184], and mustard [185].

### 4.3. Crop Breeding Assisted by Selenium Biofortification

A successful breeding methodology to produce Se biofortified food crops depends on the plant genotype, growing environment, soil physical and chemical properties and the soil microbial population. Breeding-assisted Se biofortification is a time-consuming long-term research endeavor which requires high inputs and resources. Successful implementation of a program to develop Se biofortified food crops requires identification of the most promising parent lines with broad genetic variations; long term crossing and back-crossing; and traits which are stable within a wide range of climate and soil environments [186].

### 4.4. Molecular and Genetic Engineering for Selenium Biofortification

Modern biotechnology-based Se biofortification facilitates Se transport, accumulation, volatilization and tolerance if toxicity arises [187]. The acceleration of Se transport in transgenic crops results in crop quality enhancement and seleno-amino acid accumulation in the edible parts of food crop which may be beneficial for human and animal health. Overexpression of genes involved in Se transportation in cell plasma membrane enhances the potentiality of Se uptake and subsequent transport within the plant [188]. For example, the overexpression of adenosine triphosphate sulfurylase (ATPS1) transgene in *Arabidopsis thaliana* results in higher organic Se accumulation in foliage but a decline in total Se accumulation [189]. However, the overexpression of ATPS1 accelerates the concentration of both organic and total Se in the leaves of mustard plants [190].

#### 4.4.1. Biofortification of Selenium through Molecular Approaches

While it is well documented that inter- and intraspecific variation in Se accumulation in plants exists, the genetics of Se accumulation have received little attention. Using recombinant inbred lines, the genetic basis of selenite tolerance of *A. thaliana* has recently been investigated. Three quantitative trait loci (QTLs) on chromosomes 1, 3 and 5 explained 24% of the variation in Se tolerance as defined by root-length inhibition and 32% of the phenotypic variation in terms of root length [191]. Several QTLs are associated with high Se accumulation in the grains and leaves of crops have been identified [192,193]. Also, QTLs influencing grain Se concentration have been identified using populations derived from crosses between bread wheat genotypes [176], a cross between an indica and a japonica rice variety [194], and an association mapping panel of rice accessions [195]. Yang et al. [196] identified four QTLs affecting grain Se concentration in a genetic mapping population derived from a cross between wild emmer wheat and tetraploid durum wheat. These occurred on chromosomes 5B, 6A and 6B. None of the causal genes underpinning the QTLs affecting Se accumulation in wheat or rice grains is currently known. Two QTLs affecting seed Se concentrations were identified using a population derived from a cross between two soybean cultivars, one of which includes a gene encoding GmSULTR2 which may facilitate Se translocation from the root to the shoot [183]. Plant cultivars with high Se concentrations in edible products can be used in marker-assisted breeding to transfer these high-Se QTLs to high-yielding low Se cultivars [167].

#### 4.4.2. Biofortification of Selenium through Genetic Engineering and Transgenics

Bio-fortification by traditional and molecular breeding has been achieved in crops and specific crop components when genetic diversity is available in the primary, secondary, or tertiary gene pools of the targeted crop. When genetic diversity is unavailable, genetic transformation is the better option. This transgenic approach is advantageous in that once a useful gene has been discovered it can be used to enhance Se accumulation in multiple crops.

Transgenic plants have been engineered with greater Se tolerance, Se accumulation or reduced Se volatilization than their non-transgenic counterparts [197]. An example of this is the SeCys methyltransferase gene of *Astragalus bisulcatus* (two-grooved poison vetch) which was introduced into *Arabidopsis thaliana* to overexpress Se-methyl SeCys and g-glutamyl methyl SeCys in plant shoots [198], which resulted in an increased accumulation of Se within the plant. Other genes have been successfully targeted by genetic engineering in the last decade, with positive outcomes for Se bio-fortification. For instance, the overexpression of the Se binding protein gene SBP1 in *Arabidopsis thaliana* enhanced the resistance of the plant to selenite via a GSH-dependent mechanism [199]. Similarly, the loss-of-function mutations in the gene APX1 coding for a cytosolic ascorbate peroxidase enzyme or the overexpression of the ethylene response factor ERF96 improved Se tolerance and accumulation in *A. thaliana* [200,201]. In the Se accumulator *Brassica juncea*, a novel SeCys methyltransferase enzyme has been identified, which is capable of methylating both homocysteine and SeCys substrates [202]. The overexpression of this enzyme in tobacco plants increased the total Se and MeSeCys accumulation in the plants (Chen et al. 2019). Another potential gene target is the NRT1.1B transporter, a member of the rice peptide transporter (PTR) family involved in nitrate transport; its overexpression in rice leads to higher SeMet accumulation in rice grain [203]. Additionally, the overexpression of genes encoding SeCys lyases or the overexpression of the At Se building protein (SBP) 1 increases a plant’s tolerance to selenate or selenite [188]. In addition to the assimilation enzymes, sulphate transporters may be potential targets of genetic engineering. When combined with functional genomics this gene technology could significantly contribute to future Se bio-fortification research [204]. The advancement of modern molecular tools and analytical techniques has progressed research which concentrates on Se bio-fortification to design future and more effective crop breeding strategies. Analytical methods include synchrotron X-ray fluorescence and X-ray absorption near edge structure spectroscopies, while molecular technologies benefit from high-speed and low-cost next-generation sequencing (NGS), and encompass oligo-directed mutagenesis, reverse breeding, RNA-directed gene-methylation, and gene editing [205].

### 4.5. Selenium Biofortification of Crops by Beneficial Microorganisms

Bio-fortification enhances the nutrient concentration of crops and can be undertaken by fertilizers, conventional plant breeding or by biotechnological approaches or by beneficial microbes. Bio-fortification of food crops increases their nutritional concentrations where the essential elements are in inadequate supply in the soil. Se bio-fortification can be performed successfully in Se-deficient soils by selecting plant species that can assimilate the micronutrient in their edible parts and thus enrich the diet of animals (including humans). Moreover, the excreta of animals fed Se-fortified plant matter again enrich the soil with Se.

Many agricultural soils are deficient in Se and hence biofortification is a well-established technique to improve Se uptake in food crops. Many microbes assist plants to take up nutrients from soils, resist abiotic stress, and improve their growth and yield. These beneficial microbes may also be exploited for bio-fortification. Dark septate fungi, mycorrhizal and root endophytic fungi, plant growth-promoting rhizobacteria (PGPR), etc. are used for Se bio-fortification [206].

Higher concentrations of bacteria colonizing a seleniferous area resulted in better root formation and consequently greater plant-Se uptake in *Brassica juncea* [207]. The bacteria population enhanced the bioavailability of Se; however, this may be species-dependent. Microbial biomass carbon, microbial biomass nitrogen, adenylate energy charge and metabolic quotient did not influence plant-Se uptake from soil [208].

The symbiotic relationship between food crops and beneficial microorganisms (BMOs) improves crop growth, micronutrient uptake and resistance to different stressors. Endomycorrhizal and ectomycorrhizal fungi, and root endophytic fungi (REF) are well-known BMOs commonly used as biofilm biofertilizers. Arbuscular mycorrhizal fungi (AMF) settle in roots of angiosperms while ectomycorrhizal fungi are found in gymnosperms. REFs generally have a vast range of host plants.

Se bio-fortification by microbes is not new. In soils rich in Se, rhizosphere bacteria helped in the formation of root hairs and Se uptake as selenite in mustard (*Brassica juncea*) was confirmed by de Souza et al. [208]. In 2015 Yasin et al. [207] reported that various combinations of BMOs increase Se concentration in *Brassica juncea* growing in a seleniferous area. Different bacteria may differently influence plant growth by mediating different rates of Se uptake from soil. Considering all the microflora which surround the roots of food crops grown in a Se-rich soil, important information may be obtained regarding the suitable bio-fortification approach to be taken. In some Se-enriched soils (e.g., 20 μg Se/g soil) the presence of soil microbes, such as adenosine triphosphate (ATP), adenylate energy charge (AEC), ATP-to-microbial biomass C and metabolic quotient, had a limited effect on Se uptake [208].

#### 4.5.1. Arbuscular Mycorrhizal Fungi (AMF) and Root Endophytic Fungi

Arbuscular mycorrhizal fungi are important in Se bio-fortification as they enhance the uptake of nutrients in host plants. The genomes of AMF encode high-affinity inorganic phosphate transporters [209]. There is a competition between the uptake of phosphate and Se accumulation that results in a decrease in the accumulation-translocation coefficients of Se. This also decreases the Se content in wheat (especially in leaves, stems, spikes and roots) when phosphate fertilizers are applied to Se-enriched crop fields [210]. Some species of *Astragalus*, namely, *A. racemosus* and *A. bisulacatus* are hyper accumulators of Se, whereas, *A. glycyphyllos* and *A. drummondii* are non-accumulators. Sulfur deficiency in these species increased Se accumulation; as well, an increase in Se supply enhanced the accumulation of sulfate in shoot and root tissue [211]. Table 5 and Table 6 illustrate AMFs and REFs that are used in Se bio-fortification in food crops.

Arbuscular mycorrhizal fungi increase Se uptake in plants. When wheat seedlings were inoculated with AMF, namely *Glomus versiform* and *Funneliformis mosseaein* in the hydroponic culture medium, the uptake of selenate and enhanced the accumulation of selenite; however, there was no effect on the uptake of SeMet [217,218]. Higher accumulation of Se was observed by up-regulation of three genes that encode sulfate transporters, i.e., TaSultr1:1, TaSultr1:3, and TaSultr2:1, in the roots of mycorrhiza. In the roots of *G. versiform* and *F. mosseae*, TaSultr1:1 gene expression was up-regulated 2.18-fold and 2.12-fold, respectively. Allium sativum L. contains diallyl disulfide, which inhibits metastasis of many cancer cells (colon, lung, and skin cancer cells) and WEHI-3 leukaemia cells [219].

#### 4.5.2. Plant Growth-Promoting Rhizobacteria (PGPR)

The interface between roots and soil is the rhizosphere, where critical interactions between beneficial microorganisms and plant roots occur. In 1980, Kloepper demonstrated the beneficial role of PGPR [220]. PGPR promote (directly or indirectly) plant growth in the rhizosphere. Mechanisms like bio-fertilization and photo-stimulation promote plant growth while reducing the use of inorganic chemical fertilizers, combating abiotic stress and reducing plant disease. The WHO recognizes some micronutrients which are essential for the proper functioning of the human body, including Se, iron (Fe) and zinc (Zn). PGPR have a significant role in the bio-fortification of Se, Fe and Zn [221].

Plant roots absorb Se in the forms of selenate, selenite or SeCys and SeMet, but cannot directly take up metal selenides or elemental Se. PGPR increase the level of Se uptake in plants, improving both animal and human health. Mustard (*Brassica juncea*) utilizes PGPR to increase Se accumulation and volatilization. Here, dimethyl selenide is the form of most volatilized Se; it is five- to seven-hundred times less toxic than elemental Se. This makes PGPR one of the most desirable tools for Se bio-fortification to improve Se concentrations in the human food chain. Endophytic bacteria identified and isolated from Se-supplemented wheat improve plant growth, bio-fortification and serve as biocontrol agents in wheat cultivation in ash-derived volcanic andisol in Southern Chile [222].

PGPR have well-established roles in enhancing Se uptake, crop productivity and stress tolerance [223]. Some PGPRB solubilizes inorganic phosphate in soils. Phosphate solubilizing bacteria are exploited for Se bio-fortification Table 7. Often stable complexes are formed by Se with clay minerals and/or strongly absorbed with aluminium (Al), iron (Fe), manganese (Mn), and hence have low bioavailability to most plants [224]. However, selenite and selenate are widely available to plants. When selenate and selenite (e.g., as Se fertilizer) are applied to agricultural land, they are reduced very fast and converted into forms which make them unavailable to plants (i.e., Se–metal ion complexes) which leads to low Se bioavailability (<10% only) for food crops. The re-solubility of Se in soils is very important for bio-fortification. PGPR function as Se-solubilizing agents [225]. Trivedi et al. [226] identified several endophytic seleno-bacteria from *Ricinus communis*; further identification at the molecular level showed that these were *Alcaligenes faecalis, Paraburkholderia megapolitana,* and *Stenotrophomonas maltophilia*. *P. megapolitana* play an important role in facilitating the growth of soybean (*Glycine max*) under drought conditions while increasing Se bio-fortification. The synergistic action between Se bio-fortification and increased drought tolerance is crucial for the cultivation of crops in arid and semi-arid regions where Se is limited.

## 5. Mechanisms to Uptake and Accumulate Selenium in Plants

### 5.1. Uptake Mechanisms

Selenium is present in Nature in both organic and inorganic forms. Organic forms are mainly SeMet and SeCys; while inorganic forms include elemental Se, selenide (Se^2−^), selenite (SeO_2_^−3^SeO_3_^2^^−^) and selenate (SeO_2_^−4^SeO_4_^2^^−^) [229]. The uptake, distribution and translocation of Se within plants are determined by plant translocation, the activity of membrane transporters, the presence of other substances, soil physical conditions (e.g., salinity and soil pH), the form and concentration of Se, and the plant species and phase of its development [230]. In comparison to selenite, selenate is more water-soluble and more common in agricultural soils [231]. In acidic soils, Se mainly exists as selenite, while in alkaline soils selenate is the dominant form of Se. Both these forms are metabolized to seleno-compounds but differ in terms of absorption and mobility within the plant [232]. The rates of plant transpiration and xylem loading determine the translocation of Se to shoot tissue [230]. Selenate is more mobile than SeMet, which is more mobile than either selenite or SeCys in wheat and canola [157]. The transport of selenate in plants is via sulfate channels and transporters [191], while phosphate transport mechanisms are responsible for the transport of selenite [232].

The existence of transporters for sulfate and selenate depends on the soil and plant nutritional status [233]. Constitutive active sulfate transporters showed less selectivity for sulfate over selenate than inducible sulfate transporters. Selectivity for Se decreases when there is a higher concentration of external sulfate [233]. The earlier study observed that selenate (a toxic analogue of sulfate) transport inside the plant of Arabidopsis thaliana via sulfate transporters (SULTR1 and SULTR1;2) [234], while elsewhere it was found that the uptake of selenate, sulfate transporter gene SULTR1;2 plays a predominant role in plant roots [235]. Arabidopsis thaliana was found resistant to selenate with lack of sulfate transporter gene SULTR1;2, but no resistance with another sulfate transporter gene SULTR1 [234,235,236]. There was an increase in Se uptake with the starvation of sulfur in *Triticum aestivum* [233]. Passive diffusion has been observed as a mechanism for selenite uptake [237], while Li et al. [232] found an active transport mechanism for selenite uptake. Terry et al. [197] observed selenite uptake without the involvement of membrane transporters. Li et al. [232] stated that in phosphorous deficiency the uptake of selenite was enhanced. This finding not only supports earlier studies indicating low uptake of selenite with increasing phosphate concentration but also indicates the role of phosphate transporters for selenite uptake.

### 5.2. Accumulation Mechanisms

During seedling growth, young leaves contain relatively high concentrations of Se [238]. The accumulation of Se usually occurs in the vacuoles of the plant cells [239,240], and through sulfate transporters, Se may be effluxed from the vacuole [241]. Depending on the concentration of Se inside the cell, plants are classified as non-accumulators, secondary-accumulators or hyper-accumulators [242]. Plants which thrive in regions which are rich in Se (i.e., >1000 mg Se/kg of soil) are thus hyper-accumulators of Se. Examples of hyper-accumulators include *Xylorhiza, Neptunia, Conopsis, Astragalus* and *Stanleya* species. Plants that accumulate between 100 and 1000 mg Se/kg soil with no toxicity symptoms are secondary-accumulators; these include *Medicago sativa*, *Camelina, Aster, Helianthus,* Broccoli, *Brassica napus* and *Brassica juncea* varieties. Plants which accumulate less than 100 mg Se/kg of soil are non-accumulators: in Se-enriched soils, these plants would not survive. Non-accumulator plants volatilize Se as dimethyl selenide (DMSe) and show retarded growth [242]. Non-accumulator plants sequester Se in vacuoles when enriched with Se [240].

## 6. Prospects of Selenium Biofortification

The importance of Se has long been recognized for human and animal life [4,14]. As a trace element, Se is critical for human health and Se deficiency impedes metabolism [243]. Generally, crops biofortified with Se are enriched with many beneficial phytochemicals, such as minerals and antioxidants [244] Figure 3. The response of plants to Se biofortification differs by crop, for example, different responses have been observed in lettuce [47], tomato [245], broccoli [246], cucumber [247] and carrot crops [248].

Selenium is closely bound to clay particles and is found in greater abundance in clay soils than in more sandy soils. Se toxicity is a result of the isomorphic substitution of sulfur for Se, which ultimately adversely affects plant growth. In general, however, food products are low in Se and benefit from Se enhancement, of which biofortification is very important [249]. Generally, vegetables have good prospects for Se biofortification [250] due to their inherent tendency to accumulate Se [251]. Biofortification has also been observed to occur naturally in Brazil, where Amazonian fruits contain sufficient Se to meet daily human requirements (0.03 to 512 μg/g) [252]. There are many costs associated with Se biofortification which impede the adoption of this process; efforts are being made to reduce the costs of biofortification. One application is to mix a mile (1%) Se solution with fertilizer before applying to the soil [253].

Several factors influence the success of Se biofortification: two major considerations are the method of Se fertilization (i.e., in-soil vs. foliar application) and the crop species. Generally, Se has a smaller role in the growth and development of taller plants [254]. Se is assimilated to Se-amino acids through sulfate channels [255]. Malformed proteins which cause toxicity may be produced as a result of excess Se [256], however, lower Se concentrations are beneficial, particularly for hyper-accumulator plant species [6]. The beneficial effects of Se biofortification depend on the rate of Se application, the accumulation rate of the plant, and its capacity for Se tolerance [48]. Malik et al. [257] reported reductions in electrolytic leakage which promoted cell antioxidative processes [200]. In salt-affected areas, a beneficial role of Se in food crops is observed which may be a result of improved plant growth and yield parameters [200], improvement in net photosynthesis and water content [258], and the stimulation of antioxidants. Under heat or drought stress, Se plays a critical role in maintaining cereal grain yields [259,260]. Se also protects plants from insect-pest and fungal attacks [261,262]. Figure 4 illustrates many ways in which Se biofortification improves human and animal health [131].

## 7. Conclusions

Se is an essential trace element for all living organisms. Se deficiency causes severe ill health effects in humans and animals. It is estimated that about one billion people globally suffer from Se deficiency. Se is linked in humans to the effective performance of the immune, endocrine and reproductive systems as well as brain function. Excessive Se uptake can contribute to hypotension, tachycardia, tremors, muscle contractions, hair loss, and skin and nail lesions. Therefore, a balanced daily dose of Se should be maintained. The recommended dosage of Se varies with age, gender, pregnancy, lactation and also geographical location and diet. Se is accrued in the human body through the consumption of animal and plant foods. To meet Se deficiency and improve human health where Se is inadequate, Se concentrations in plants should be quantified and balanced. Se-biofortification in combination with agronomic, breeding, molecular, biotechnology, and genetic engineering management approaches are effective strategies for Se enrichment in human plants foods.

## Figures and Tables

**Figure 1 molecules-26-00881-f001:**
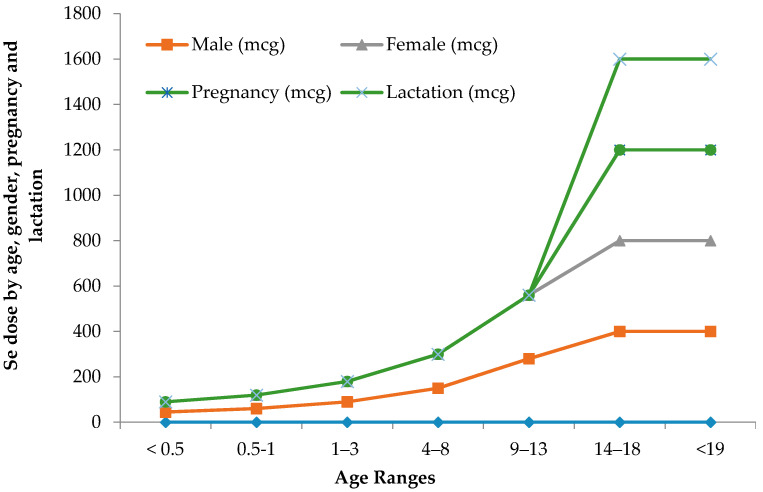
Recommended maximum daily dietary allowances for Se.

**Figure 2 molecules-26-00881-f002:**
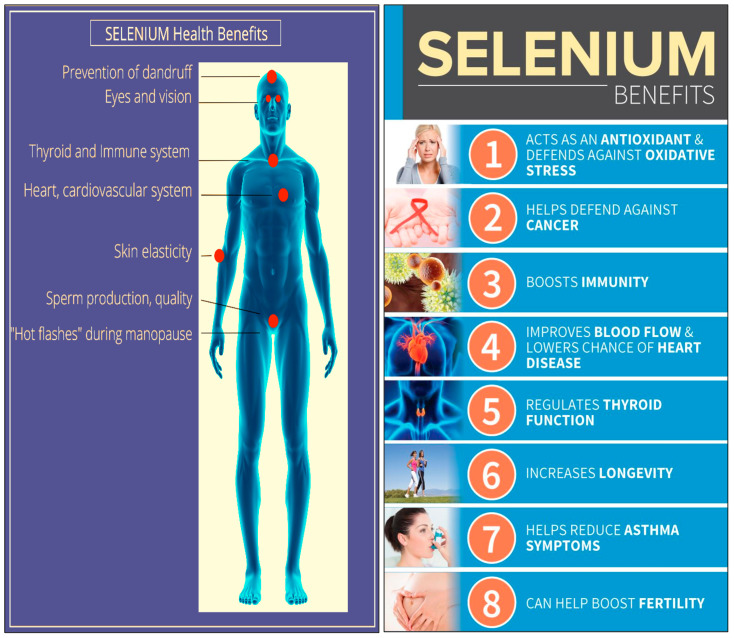
Health benefits of Se in the physiological processes of the human body.

**Figure 3 molecules-26-00881-f003:**
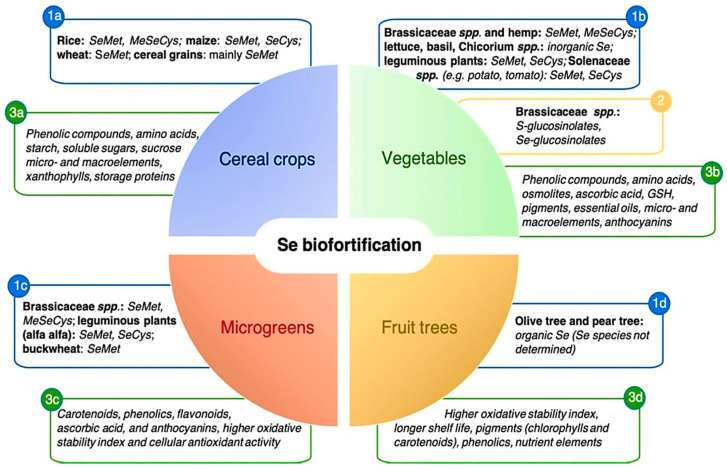
Se biofortification to improve human plant-foods [244]. Note: (1a), (1b), (1c) and (1d) indicate that SeMet, MeSeCys, SeCys contents are available in cereals, vegetables, fruits, and microgreens; (2) indicate that S-glucosinolate and Se- glucosinolate content are available only in vegetables and (3a), (3b), (3c) and (3d) indicate that phenolic compounds, amino acids, starch, soluble sugars, sucrose, micro-and macro-elements, antioxidants, storage protein etc. are available in cereals, vegetables, fruits, and microgreens.

**Figure 4 molecules-26-00881-f004:**
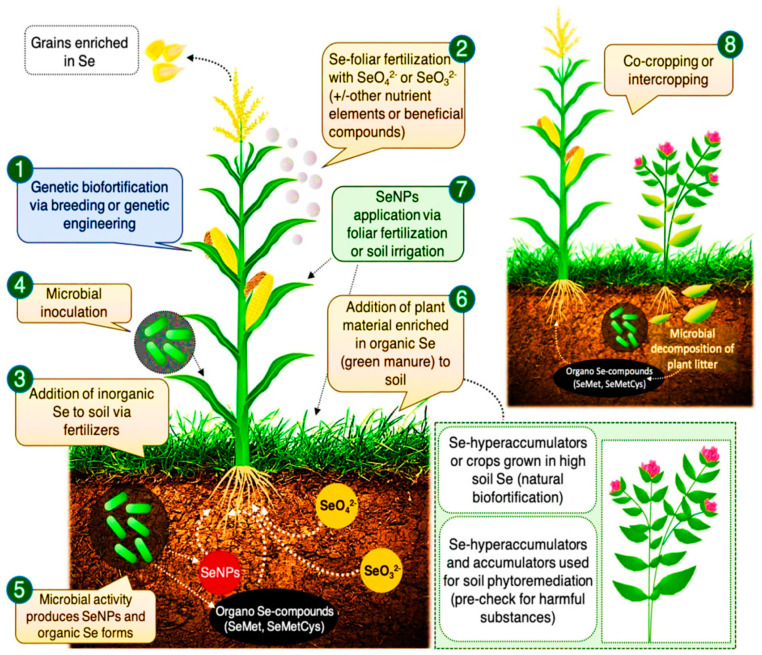
Se biofortification promotes crop yields and quality parameters. Se-biofortification approaches include (**1**) genetic tools, (**2**) through foliar application, (**3**) soil amendment, (**4**) agronomic biofortification, (**5**) broadcasting into soils, (**6**) green manure with Se, enriched growth and development of plants, (**7**) nano-sized biofortification to leaves or soil, and (**8**) intercropping with Se- hyper-accumulator plants. Adapted and modified from Schiavon et al. [132].

**Table 1 molecules-26-00881-t001:** Se responsive diseases in animals.

Syndrome	Clinical Features
White Muscle Disease	Acute onset, stiffness, skeletal or cardiac muscles affected.
Reproductive performance	The retained fetal membrane in dairy cows.
Abortion, Still-births	Late third trimester abortions and stillbirths
Myodegeneration of cattle (adult)	Myocardial fibrosis, myoglobinuria weakness
Infertility in cattle and sheep	Decreased conception rate, early embryonic death
Diarrhoea	Diarrhoea, weight loss in young and adult cattle

Source: Information in Table 1 was collected from Gupta and Gupta [32] with permission.

**Table 2 molecules-26-00881-t002:** Recommended maximum daily Se intake levels.

Age	Male	Female	Pregnancy	Lactation
Birth to 6 months	15 mcg *	15 mcg *		
7–12 months	20 mcg *	20 mcg *		
1–3 years	20 mcg	20 mcg		
4–8 years	30 mcg	30 mcg		
9–13 years	40 mcg	40 mcg		
14–18 years	55 mcg	55 mcg	60 mcg	70 mcg
19–50 years	55 mcg	55 mcg	60 mcg	70 mcg
51+ years	55 mcg	55 mcg		

Source: The information in Table 2 is collected from IMFNB [41] with permission. * Breast milk, formula, and food should be the only sources of selenium for infants.

**Table 3 molecules-26-00881-t003:** Results of a Se-biofortification program in Finland.

Years	Case Study	References
1970	East Karelia has the highest heart disease rates in the world	Aro et al. [89]
	Low available Se in soils.	
	Se supplementation of livestock feeds commences	
	Heart disease (especially in men) begins to decline	
1984	National Se biofortification program commences	Broadley et al. [87]
1987	Se in spring wheat grain increases from 10 (pre-1984) to 250 µg/kg	Eurola et al. [90].
	Human Se intake trebles	
	Human plasma Se level doubles (55 to 107 µg/kg)	Broadley et al. [88].
	Heart disease continues to decline	
2010	Heart disease relatively low (resulting from reduced smoking, improved diet and exercise, and possibly higher Se status)	Mäkelä et al. [91]
	No detrimental effects of Se observed	Varo et al. [92]
	Se still added to crop fertilizers at 10 mg/kg	

**Table 5 molecules-26-00881-t005:** Arbuscular mycorrhizal fungi used in Se bio-fortification.

Host Plants	AMF	References
*Allium sativum*	*Glomus fasciculatum*	Patharajan and Raaman [212]
*Allium sativum*	*Glomus irtraradices*	Larsen et al. [205,213]
*Lolium perenne, Allium sativum, Medicago sativa, Glycine max, Zea mays*	*Glomus mosseae*	Patharajan and Raaman [212]; Yu et al. [214]
*Glomus versiform*	*Triticum aestivum*	Luo et al. [215]
*Lactuca sativa, Asparagus o* *fficinalis, Lactuca sativa, Allium cepa*	*Rhizophagus intraradices*	Sanmartin et al. [216]

**Table 6 molecules-26-00881-t006:** Root endophytic fungi used in Se bio-fortification.

Host Plants	Root Endophytic Fungi	References
*Stanleya pinnata*	*Alternaria seleniiphila*	Lindblom et al. [211]
*Astragalus bisulcatus*	*Alternaria astragali*	
*Stanleya pinnata*	*Aspergillus leporis*	
*Astragalus racemosus*	*Fusarium acuminatum*	
*Allium cepa*	*Trichoderma harzianum*	Sanmartin et al. [216]

**Table 7 molecules-26-00881-t007:** Plant growth-promoting rhizobacteria (PGPR) which facilitate the uptake of Se.

Host Plants	PGPRB	References
*Triticum aestivum*	*Acinetobacter* sp.	Durán et al. [227]
*Ricinus communis, Glycine max*	*Alcaligenes faecalis*	Trivedi et al. [226]
*Triticum aestivum*	*Anabaena* sp.	Abadin et al. [225]
*Arabidopsis thaliana*	*Bacillus amyloliquefaciens*	Wang et al. [228]
*Triticum aestivum*	*Bacillus axarquiens*	Durán et al. [227]
*Triticum aestivum*	*Bacillus cereus*	Yasin et al. [207]

## Data Availability

Not applicable.

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
