# Peer review of "Selenium Biofortification: Roles, Mechanisms, Responses and Prospects"

_molecules, 2021, doi:10.3390/molecules26040881_

Round 1
Reviewer 1 Report
The MS is well summarising the known facts about the possibilities of Se biofortification.
It can be clearly seen that the MS was written by multiple authors, it contains a lot of facts, sentences even paragraphs that are repeatedly occurring among the whole MS. There are many repetitive statements that can be removed and instead, the authors should focus more on key messages.
The MS is easy to read, it is a very book chapter like, in my opinion, it is a summary of the current knowledge, but a review should be more critical. I miss the highlight of the future aspects, the specific suggestions besides the general findings. This is a very wide field, I know, but authors should try to give something new, that can further support the science, to create new ideas, and to try to solve the given problem.
Author Response
Response to Reviewer_1 comments
Comments and Suggestions for Authors
The MS is well summarising the known facts about the possibilities of Se biofortification.
It can be clearly seen that the MS was written by multiple authors, it contains a lot of facts, sentences even paragraphs that are repeatedly occurring among the whole MS. There are many repetitive statements that can be removed and instead, the authors should focus more on key messages.
The MS is easy to read, it is a very book chapter like, in my opinion, it is a summary of the current knowledge, but a review should be more critical. I miss the highlight of the future aspects, the specific suggestions besides the general findings. This is a very wide field, I know, but authors should try to give something new, that can further support the science, to create new ideas, and to try to solve the given problem.
Authors’ response: Thank you so much for your good suggestion. We have thoroughly checked the whole manuscript and have been removed all repeated facts/sentences. We also highlighted the importance overview discussed inside the review. I am sure you will be now happy to read the present form of the review. All edits are showing in track change mode in the text of the article.
Reviewer 2 Report
The manuscript 'Selenium biofortification: role, responses, prospects and mechanisms' is a nice review and could be published after checking a few issues:
Major concerns
1) The review highlights from the beginning the benefits of supplementation and the troubles caused by the deficiencies. But the information about the toxicity of selenium and the consequences of an excess of it is dispersed along the manuscript (very short at the introduction, and sections 6 and 8, which are not exactly the issue. It would be advisable, to have a new section "Selenium toxicity" at the beginning of the article, just after the selenium deficiency section, to provide a quicker insight of the two aspects, which are critical.
Minor concerns
2) The list of affiliations could be shortened to have in the same line all the authors of the same Department (with all mails later in the same line), and the titles/position of each author is not necessary. This would avoid having the first page only for the affiliations, when several of them are repeated.
3) Typos or incorrections (please check manuscript carefully to correct additional ones):
- Abstract, line 42. SeCys and SeMet are not transformed to seleno-proteins. The correct word would be "are incorporated".
- Line 158: please write 21st Century with superscript.
- 'Valance' in line 450 must be changed by 'Valence'
- Uptake is repeated twice in line 615.
4) Please cite tables 1 and 2 in the text before its first appearance: currently tables are cited after. Then, tables need to be moved down as necessary to have the citation first.
5) Please avoid splitting a table between 2 pages, as happens with Table 2.
6) Please add references/sources to table 1, 2 and 8. Also to Figures 1-4 if necessary; and to any other table I may have missed. This will also eliminate potential lack of consistency. For example, the daily intake for adults in Table 2 is 55-70 micrograms whereas in line 167 is 55-200 micrograms.
7) As a general concern about tables, references in the majority of them are put like a list (examples, but not limited to; Tables 3 and 4). Please write each reference in line with its corresponding table line. It is no problem to need to enlarge the tables if any needs to be repeated.
8) In Table 4, for it to have better quality of presentation, I would recommend using all the page width (including the left margin), to avoid the need to split words between two lines.
9) Line 355-356. For completion issues, it would be interesting to mention which ions favour selenate absorption and which ones difficult it.Author Response
Reviewer-2
Response to Reviewer_2 comments
Comments and Suggestions for Authors
The manuscript 'Selenium biofortification: role, responses, prospects and mechanisms' is a nice review and could be published after checking a few issues:
Major concerns
1) The review highlights from the beginning the benefits of supplementation and the troubles caused by the deficiencies. But the information about the toxicity of selenium and the consequences of an excess of it is dispersed along the manuscript (very short at the introduction, and sections 6 and 8, which are not exactly the issue. It would be advisable, to have a new section "Selenium toxicity" at the beginning of the article, just after the selenium deficiency section, to provide a quicker insight of the two aspects, which are critical.
Authors’ response: Many thanks for your good suggestions, which have allowed further to improve the quality of the review. As per your suggestion, a section on Se toxicity in humans, animals and in plants are added after the section Se deficiency.
Minor concerns
2) The list of affiliations could be shortened to have in the same line all the authors of the same Department (with all mails later in the same line), and the titles/position of each author is not necessary. This would avoid having the first page only for the affiliations, when several of them are repeated.
Authors’ response: Thank you so much for the informative suggestion, which help us to reduce the volume of the review.
3) Typos or incorrections (please check manuscript carefully to correct additional ones):
- Abstract, line 42. SeCys and SeMet are not transformed to seleno-proteins. The correct word would be "are incorporated".
Authors’ response: Abstract has been rewritten as per your suggestion
- Line 158: please write 21st Century with superscript.
Authors’ response: The correction has been made
- 'Valance' in line 450 must be changed by 'Valence'
Authors’ response: The suggested edit has been done
- Uptake is repeated twice in line 615.
Authors’ response: The corrections have been done. Please check all edits as track change mode.
4) Please cite tables 1 and 2 in the text before its first appearance: currently tables are cited after. Then, tables need to be moved down as necessary to have the citation first.
Authors’ response: We have now cited both Tables (1 & 2) closely to the text.
5) Please avoid splitting a table between 2 pages, as happens with Table 2.
Authors’ response: As per your suggestion, we have checked all Tables and Figures and avoided to split between two pages
6) Please add references/sources to table 1, 2 and 8. Also to Figures 1-4 if necessary; and to any other table I may have missed. This will also eliminate potential lack of consistency. For example, the daily intake for adults in Table 2 is 55-70 micrograms whereas in line 167 is 55-200 micrograms.
Authors’ response: We have now added sources for Table 1, 2 and 8
7) As a general concern about tables, references in the majority of them are put like a list (examples, but not limited to; Tables 3 and 4). Please write each reference in line with its corresponding table line. It is no problem to need to enlarge the tables if any needs to be repeated.
Authors’ response: As per your suggestion we have added sources of information in the line of Tables 3 & 4
8) In Table 4, for it to have better quality of presentation, I would recommend using all the page width (including the left margin), to avoid the need to split words between two lines.
Authors’ response: For better presentation width of Table 4 is enlarged and also References are presented in lines of Table 4.
9) Line 355-356. For completion issues, it would be interesting to mention which ions favour selenate absorption and which ones difficult it.
Authors’ response: As per the references of [124-126], in the case of alkaline soils, the dominant form of Se is selenate. While absorption of selenate by plants is sometimes affected mainly due to the presence of ions K+, Ca2+, Mg2+
N.B. All edits are showing in track change mode in the text of the article.
Reviewer 3 Report
The review entitled “Selenium biofortification: role, responses, prospects and mechanisms” summarizes the roles, responses, and mechanisms of selenium (Se) in humans and the potential to enrich plant foods with this micronutrient through biofortification. Overall, it is a novel work, well written and with possible impact that is relevant for crop and food production that could potentially benefit human health. However, I have a couple of major concerns that worry me for the potential publication of this manuscript:
- Errors in the citations: For example, reference 218 (Teotia et al., 2016) refers to potassium and not to Se. The authors of this work are from India and not from Chile. In this article there is no mention of Se or Chile, so this must be rectified.
Reference 259 (D´Amato et al., 2018) deals with Se supplementation in olive grown under water stress, and Reference 260 (Jotwiak and Politycka, 2019) refers to cucumbers trialed under water deficit. These two references are not related to cereals as the present form of the manuscript indicates. Another mistake is that Schiavon et al. does not correspond to reference 163, but to reference 177 (see line 767). These examples of quotation errors put the validity and scientific accuracy of this work at risk, which would make it advisable to fully review the 266 references included in this article. - Length of the article: It is very long and several ideas are repeated; I recommend to cut the extent of this article in 30%. For instance, section 8 (Se phytotoxicity) (lines 770-783) reiterates previous concepts that could be merged into the section 3.2.1. Other repetitions of different concepts and ideas are stated below. In addition, I would also reduce the number of figures and tables, and improve them based on my comments below.
Other comments and remarks are the following:
- I suggest to change the article´s title to “Selenium biofortification: roles, mechanisms, responses and prospects”.
- Transfer lines 167-168 to line 81.
- Ideas contained on lines 78 and 176 are repeated. Eliminate either one.
- Table 2 is not very informative. I recommend delete it and use words instead.
- Line 252: Do you mean “multiple-nutrient fertilizer mix”?
- Table 4: How come you have very high Se content in soils associated to low levels of Se in cereal grains in East Zimbabwe?
- Lines 330-335: This looks very general. Please quantify the doses of Se used giving specific examples.
- Lines 342-344: What are considered high Se concentrations? Are soil analyses routinely carried out to determine the soil Se availability? Please refer to the technical difficulties to specifically quantify the soil Se levels.
- Lines 346-351: The authors need to quantify the Se amounts by giving ranges.
- Line 359: Which ones?
- Lines 380-381: What do the authors mean with high temperatures and high soil pH? Please specify.
- Line 394: At late vegetative phenological stages?
- Lines 395-398: Is there any technical advantage of using soil fertilization relative to Se foliar application?
- Line 406: How higher?
- Line 455: Which are the constraints of organic fertilization?
- The paragraph containing lines 468-477 is not related to traditional breeding approach. I suggest to put it in the section 4.1 (Biofortification through agronomic management).
- Section 4.2 refers to genetic variation of Se in different plant species, but not to traditional breeding approach that involves directed crosses followed by several generations of phenotypic selection to generate a new plant cultivar high in Se. I recommend to re-name this section and/or explain hoy is traditional breeding specifically conducted to increase Se levels on plants.
- Line 484: Use “broad” instead of “good”.
- Lines 518-512: Is it worth to do conventional and MAB plant breeding to increase Se content? Although I recognize the value of this section, I think that conventional breeding to increase Se content is not an important breeding trait since there are other characters that have higher priority on the breeding programs, so I suggest to shorten this section.
- Line 557: Utilize “gene” instead of “DNA”.
- Line 585: Indicate the reference number for Souza et al. 1999.
- Line 595: Which are the important factors affecting Se uptake?
- Tables 5 and 6: The relationships between the host plants, fungi and the references are unclear. Please use well defined and sharp rows to differentiate them that will improve the quality of these tables.
- Line 684: What do the authors mean with “selenate resistance”? Need to define this term.
- Line 709: Toxic for plants, animals or humans?
- Line 722: Add “daily human dietary”.
- Line 723: Add “Se intake levels of humans”.
- Figure 3: Unclear what the different colors and numbers (i.e., 1a, 1b, 3a, 3d) stand for in this figure. Please explain this ambiguity on the legend.
- Line 740: The authors repeat the same idea than in line 67. Delete either one.
- Line 794: Use “should be quantified and balanced” rather than “improved”
Author Response
Reviewer-3
Response to Reviewer 3 comments
Comments and Suggestions for Authors
The review entitled “Selenium biofortification: role, responses, prospects and mechanisms” summarizes the roles, responses, and mechanisms of selenium (Se) in humans and the potential to enrich plant foods with this micronutrient through biofortification. Overall, it is a novel work, well written and with possible impact that is relevant for crop and food production that could potentially benefit human health. However, I have a couple of major concerns that worry me for the potential publication of this manuscript:
- Errors in the citations: For example, reference 218 (Teotia et al., 2016) refers to potassium and not to Se. The authors of this work are from India and not from Chile. In this article there is no mention of Se or Chile, so this must be rectified.
Reference 259 (D´Amato et al., 2018) deals with Se supplementation in olive grown under water stress, and Reference 260 (Jotwiak and Politycka, 2019) refers to cucumbers trialed under water deficit. These two references are not related to cereals as the present form of the manuscript indicates. Another mistake is that Schiavon et al. does not correspond to reference 163, but to reference 177 (see line 767). These examples of quotation errors put the validity and scientific accuracy of this work at risk, which would make it advisable to fully review the 266 references included in this article.
Authors’ response: Thank you so much for your valuable comments. As per your suggestion, we have thoroughly checked all references and corrected where necessary. For example, Teotia et al., 2016 replaced by Durán et al., 2014; D´Amato et al., 2018 replaced by Manojlović et al., 2019 and Jotwiak and Politycka, 2019 replaced by Sarwar et al. 2020. In line 767 and Fig. 4, Reference [163] has been changed to [177].
- Length of the article: It is very long and several ideas are repeated; I recommend to cut the extent of this article in 30%. For instance, section 8 (Se phytotoxicity) (lines 770-783) reiterates previous concepts that could be merged into the section 3.2.1. Other repetitions of different concepts and ideas are stated below. In addition, I would also reduce the number of figures and tables, and improve them based on my comments below.
Authors’ response: Thanks for your good suggestion. As per your and second reviewer suggestion, we have minimized the repeated ideas through adding a section (2.2 Selenium toxicity; 2.3.1. Toxicity in humans and animals; 2.3.2 Selenium phytotoxicity) about phytotoxicity of Se in humans and plants.
Other comments and remarks are the following:
- I suggest to change the article´s title to “Selenium biofortification: roles, mechanisms, responses and prospects”.
Authors’ response: As per your suggestion we have been edited the title as: ‘Selenium biofortification: roles, mechanisms, responses and prospects’.
- Transfer lines 167-168 to line 81.
Authors’ response: We have transferred the sentence to the section: 2.3.1, now line 178
- Ideas contained on lines 78 and 176 are repeated. Eliminate either one.
Authors’ response: We have been modified the information and replaced in the appropriate place.
- Table 2 is not very informative. I recommend delete it and use words instead.
Authors’ response: We agreed with you and removed the Table 2
- Line 252: Do you mean “multiple-nutrient fertilizer mix”?
Authors’ response: Yes, we have now modified these words as multiple-nutrient fertilizer mix
- Table 4: How come you have very high Se content in soils associated to low levels of Se in cereal grains in East Zimbabwe?
Authors’ response: British Geographical Survey conducted by Fordyce et al. [109], indicated that soils of Noth-East Zimbabwe are high-concentration of several elements including Se also, due to parent material
- Lines 330-335: This looks very general. Please quantify the doses of Se used giving specific examples.
Authors’ response: We have discussed details in sub-sections 4.1.1-4.1.3
- Lines 342-344: What are considered high Se concentrations? Are soil analyses routinely carried out to determine the soil Se availability? Please refer to the technical difficulties to specifically quantify the soil Se levels.
Authors’ response: In the book: ‘Biofortification: improving the nutritional quality of staple crops’ Winkler (2011) revealed that applying Se broadly at high concentrations is generally not economically sustainable, and so site-specific Se applications which account for existing Se available within the soil and crop demand must be considered. You will agree with the discussion and also it is a universal truth that excessive and imbalance use of nutrients including Se is not economically viable and environmentally friendly.
- Lines 346-351: The authors need to quantify the Se amounts by giving ranges.
Authors’ response: As per your suggestion, we have added an example, conducted by Ramkissoon et al. (2019), who revealed that the valuable pathway to increase Se in the human diet, thus preventing Se deficiency by increasing the Se concentration of staple crops through fertilization. They observed that application of 3.33 µg kg−1 of Se (equivalent to 10 g ha−1) to wheat can be made more efficient by its co-application with macronutrient carriers, either to the soil or to the leaves during the awn-peep stage and observed that grain Se concentrations varying from 0.13–0.84 mg kg−1. For raising the grain Se concentrations, soil application of selenate was found 2–15 times more effective than granular Se-enriched micronutrient fertilizers. While co-application of Se as foliar with an N carrier doubled the Se concentration in wheat grains compared to the application of foliar Se.
- Line 359: Which ones?
Authors’ response: Absorption of Se within soils may be reduced due to the presence of competitive ions such as K+, Ca2+, Mg2+, SO2−4 and Cl−
- Lines 380-381: What do the authors mean with high temperatures and high soil pH? Please specify.
Authors’ response: We have modified these sentences as: The form of selenates are highly water-soluble and available to uptake by plants, while easily leached from the soil solution. In aerated soils with neutral to higher pH, this form of Se are dominated. For example, soils with a high content of Ca and Mg CaSeO4, generally creates MgSeO4, this form is easily soluble and represent total Se soluble in a soil [132]. In soils rich in organic matter and water and without air entry selenates are transformed and reduced to less mobile forms. With decreasing pH and redox potential in soil SeO32- dominate, being less available for plants than SeO42-[133].
- Line 394, 395-398:: At late vegetative phenological stages? Is there any technical advantage of using soil fertilization relative to Se foliar application?
Authors’ response: We have been modified this paragraph as:
During foliar application, the Se solution must be distributed using well-calibrated equipment; spraying should not occur on rainy or windy days, and applications need to be made at the late vegetative stage, where there is an adequate surface area to facilitate maximum absorption of Se. In the soil application, Se is effective in the period from early growth of seedlings to plant maturity for uptaking Se by roots [140].
- Line 406: How higher?
Authors’ response: Thank you so much for your good comment. We have added details findings in your mentioned place. For example, a field study with the purple-grained wheat cultivar (202w17) and common wheat cultivar (Shannong 129) showed that both soil and foliar application of Se boosted the organic Se concentration in roots, shoots and grains of both cultivars, but the higher concentration of Se in the grain of two cultivars was noted when Se was applied as foliar. Foliar application of Se enhanced approximately 1.5-fold higher concentration of organic Se in grains of cultivar 202w17 than cultivar Shannong [140].
- Line 455: Which are the constraints of organic fertilization?
Authors’ response: In the previous version of the article, line 450-455 and present version of the article line 501-506, we revealed that Se accumulation in plants is higher when applied by mixing with organic compounds than the inorganic forms of Se [145]. For example, seleno-amino acids are active compounds and can be applied to the soil in Se-amended organic manures in crop fields [160]. Also, organo-Se compounds can also be released in soils through the decomposition of plant material and soil microbial matter [161].
- The paragraph containing lines 468-477 is not related to traditional breeding approach. I suggest to put it in the section 4.1 (Biofortification through agronomic management).
Authors’ response: Thank you so much for your good suggestion, we have been placed this paragraph now in section 4.1.
- Section 4.2 refers to genetic variation of Se in different plant species, but not to traditional breeding approach that involves directed crosses followed by several generations of phenotypic selection to generate a new plant cultivar high in Se. I recommend to re-name this section and/or explain hoy is traditional breeding specifically conducted to increase Se levels on plants.
Authors’ response: As per your suggestion we rename the section; 4.2 Success of Se-biofortification in food crops depends on a better understanding of the genetic variation of crop cultivars
- Line 484: Use “broad” instead of “good”.
Authors’ response: Thanks, we have been replaced the word ‘good’ by ‘broad’. Now line 532
- Lines 518-512: Is it worth to do conventional and MAB plant breeding to increase Se content? Although I recognize the value of this section, I think that conventional breeding to increase Se content is not an important breeding trait since there are other characters that have higher priority on the breeding programs, so I suggest to shorten this section.
Authors’ response: As per your suggestion, we have been removed the unnecessary sentences.
- Line 557: Utilize “gene” instead of “DNA”.
Authors’ response: The edit has been done
- Line 585: Indicate the reference number for Souza et al. 1999.
Authors’ response: We have added the reference in the list as:
de Souza, M.P., Chu, D., Zhao, M., Zayed, A.M., Ruzin, S.E., Schichnes, D., Terry, N. Rhizosphere bacteria enhance selenium accumulation and volatilization by indian mustard. Plant Physiology, 1999, 119(2):565-74. doi: 10.1104/pp.119.2.565.
- Line 595: Which are the important factors affecting Se uptake?
Authors’ response: Yasin, et al. (2015) observed that soils with enrichment of Se (e.g. 20 μg Se/g soil) and the presence of soil microbes, such as adenosine triphosphate (ATP), adenylate energy charge (AEC), ATP-to-microbial biomass C and metabolic quotient, had a limited effect on Se uptake.
- Tables 5 and 6: The relationships between the host plants, fungi and the references are unclear. Please use well defined and sharp rows to differentiate them that will improve the quality of these tables.
Authors’ response: Table 5 is in good condition with the line of reference(s) for each row in the Table. In the case of Table 6, the information in the first 4 rows was collected from the study of Lindblom et al. [213] and next row’s information was from the study of Sanmartin et al. [209]
- Line 684: What do the authors mean with “selenate resistance”? Need to define this term.
Authors’ response: Earlier study observed that selenate (a toxic analogue of sulfate) transport inside the plant of Arabidopsis thaliana via sulfate transporters (SULTR1 and SULTR1;2)[229], while elsewhere it was found that the uptake of selenate, sulfate transporter gene SULTR1;2 plays a predominant role in plant roots [230]. Arabidopsis thaliana was found resistant to selenate with the lacking of sulfate transporter gene SULTR1;2, but no resistance with another sulfate transporter gene SULTR1[229, 230, 231].
- Line 709: Toxic for plants, animals or humans?
Authors’ response: Both excessive and insufficiency of Se are detrimental to plants, animals or humans health. To know details about Se toxicity in humans, animals and plants, we have added a section 2.3
- Line 722: Add “daily human dietary”.
Authors’ response: Please check section 2.3 to know details. The recommended dietary allowance of Se varies in humans with age, gender, pregnancy and lactation (Fig. 1). Pregnant or lactating women require higher 9% and 27% daily amounts of Se than other women [241]. Human beings must consume around 55 micrograms of Se per day and not exceed the maximum limit of 400 micrograms per day [44]. The World Health Organization recommends a daily average intake of 55 µg of Se, while the recommendation is varied with age, gender, diet and geographic location [10]. The International Food and Nutrition Board has recommended an average daily intake of 40-70 µg Se and 45-55 µg Se for men and women, respectively, and 25 µg Se for children [2,11]. A daily dose of 55-200 µg of Se is recommended for healthy adult humans [42,43].
- Line 723: Add “Se intake levels of humans”.
Authors’ response: Please read the previous response
- Figure 3: Unclear what the different colors and numbers (i.e., 1a, 1b, 3a, 3d) stand for in this figure. Please explain this ambiguity on the legend.
Authors’ response: Note: 1a, 1b, 1c and 1d indicate that SeMet, MeSeCys, SeCys contents are available in cereals, vegetables, fruits and microgreens; 2 indicate that S-glucosinolate and Se- glucosinolate content is available only in vegetables and 3a, 3b, 3c and 3d indicate that phenolic compounds, amino acids, starch, soluble sugars, sucrose, micro-and macro-elements, antioxidants, storage protein etc. are available in cereals, vegetables, fruits and microgreens.
- Line 740: The authors repeat the same idea than in line 67. Delete either one.
Authors’ response: We have removed from here
- Line 794: Use “should be quantified and balanced” rather than “improved”
Authors’ response: The suggested edit has been done
N.B. All edits are showing in track change mode in the text of the article.
Reviewer 4 Report
The group of authors presented an interesting review of selenium biofortification. I have no expertise to comment about the human health sections. The agricultural section was properly presented. I have three suggestions.
1.) The paper is very lengthy. Depending on the requirement of the journal, I suggest the authors to cut down significant part of text and low quality references.
2.) The figures appears to be adapted from other papers. Please ensure that the copyright permission is obtained by the authors.
3.) Another figure (Figure 5) should be drawn by the authors to summarize the whole paper (biofortification, crop production and human health).
Author Response
Reviewer-4
Response to Reviewer_4 comments
Comments and Suggestions for Authors
The group of authors presented an interesting review of selenium biofortification. I have no expertise to comment about the human health sections. The agricultural section was properly presented. I have three suggestions.
Authors’ response: Thank you so much for your informative suggestions.
1.) The paper is very lengthy. Depending on the requirement of the journal, I suggest the authors to cut down significant part of text and low quality references.
Authors’ response: As per your suggestion, we have rearranged several sections of the review and also have been tried to repeated sentences for minimizing the volume of the article.
2.) The figures appears to be adapted from other papers. Please ensure that the copyright permission is obtained by the authors.
Authors’ response: Figures used in the review have been adapted and modified with permission from the original sources.
3.) Another figure (Figure 5) should be drawn by the authors to summarize the whole paper (biofortification, crop production and human health).
Authors’ response: Thank you so much for your good suggestion. We have already added Fig. 2, which discussed on human health benefits of Se, Fig. 3 highlights on Se biofortification to improve human plant-foods and Fig. 4. Focused on Se-biofortification approaches to enrich Se in foods.
N.B. All edits are showing in track change mode in the text of the article.
Round 2
Reviewer 1 Report
The MS has been improved since I saw it last time.
It is a pleasure to see that there much fewer repetitions in the text.
Unfortunately, some of the new paragraphs are not meaningful, should be checked again (see lines 185-188 and 648-651).
Otherwise, it is still looking like a book chapter, I still cannot find any future perspectives in it.
Reviewer 3 Report
The manuscript was properly improved and the present form is close to publication.